# Time-Based Corporate-Social-Responsibility Evaluation Model Taking Chinese Listed Forestry Companies as an Example

**Xinfei Li [1], Baodong Cheng [1] and Heng Xu [2],***

[1] School of Economics and Management, Beijing Forestry University, Beijing 100083, China; lxfbjfu@163.com (X.L.); baodongcheng@163.com (B.C.)

[2] Business School, China University of Political Science and Law, Beijing 100088, China

* Correspondence: hengxu@cupl.edu.cn

**Abstract:** With the rapid development of the economy, corporate social responsibility (CSR) is receiving increasing attention from companies themselves, but also increasing attention from society as a whole. How to reasonably evaluate the performance of CSR is a current research hotspot. Existing corporate-social-responsibility evaluation methods mostly focus on the static evaluation of enterprises in the industry, and do not take the time factor into account, which cannot reflect the performance of long-term CSR. On this basis, this article proposes a time-based entropy method that can evaluate long-term changes in CSR. Studies have shown that the completion of CSR in a static state does not necessarily reflect the dynamic and increasing trend of CSR in the long term. Therefore, the assessment of CSR should consider both the static and dynamic aspects of a company. In addition, the research provides the focus of different types of forestry enterprises in fulfilling CSR in the long term, and provides a clearer information path for the standard identification and normative constraints of different types of forestry enterprises CSR.

**Keywords:** corporate social responsibility (CSR); entropy approach; forestry; time factors

## 1. Background and Introduction

Corporate social responsibility (CSR) refers to the welfare-spillover behavior that enterprises should undertake, targeting shareholders, consumers, governments, the environment, and communities on the basis of profit maximization [1]. It is an important behavior of stakeholder relationship construction [2], an increasingly important strategic issue in the management field, and a research focus and hotspot in the academic field [3]. With the continuous optimization of corporate governance structures in recent years, corporate social responsibility has received extensive attention from academia and society [4]. As far as enterprises are concerned, corporate social responsibility is regarded as an important factor for enterprises to enhance their own market competitiveness [5,6]. Many multinational companies place corporate social responsibility at the same important position as those of the pricing and quality of products or services. By integrating social responsibility into the company's business process, enterprises can create a good corporate culture and brand image, fully integrate internal and external resources, create a good competitive environment, and obtain greater competitive advantages [7]. At the same time, consumers incorporate corporate social responsibility into the evaluation and decision-making process [8]. Numerous studies showed that the implementation of corporate social responsibility makes consumers evaluate a company higher [9–11]. Given the competitive conditions of the market and companies, consumers prefer to choose companies that adhere to social responsibilities [12,13]. In summary, corporate-social-responsibility behaviors can help companies better develop by establishing a good company reputation and gaining consumer recognition [14,15].

In summary, the evaluation of corporate social responsibility is of great significance to the future development of a company. First, by evaluating the degree of corporate-social-responsibility fulfillment, it is possible to better clarify the lack of corporate social responsibility, and guide the company to make adjustments in the process of fulfilling social responsibility in the future. Second, the evaluation of social responsibility can rank similar enterprises at a unified level, and supervise the revision of the performance of corporate social responsibility within a certain period of time.

Listed forestry companies take forest resources and their products as their business objects. Because of the forest resources they own, they need to assume special social responsibilities that are different from those of other industries. They are protecting the ecological environment, conserving water sources, maintaining biodiversity, and low carbon. This plays an important role in the development of the emission-reduction economy [16]. In April 2018, the Ministry of Commerce of China issued the Enterprise Green Procurement Guidelines, which, to a certain extent, urge and encourage upstream companies to improve environmental performance and influence, and pass on end consumers to establish a green lifestyle. As the main microeconomic body of forestry industry development, listed forestry companies can create huge ecological value, realize ecological safety, and improve the ecological environment with their business activities. The issue of fulfilling their social responsibilities has attracted increasing attention from academic circles. However, compared with research on the social responsibility of other types of enterprises, there is less research on the social responsibility of forestry enterprises. Forestry companies rely on forest resources [17], and forestry has greater impact on the environment or society than that of general industries. Forestry corporate social responsibility plays a key role in global sustainable development [18]. At the same time, because the production and operation of forestry companies directly impact the natural environment, forestry companies can easily become the object of public criticism [19]. Therefore, corporate social responsibility is a means for forestry companies to respond to various challenges and minimize various risks. The operation of forestry companies must be consistent with sustainable-development goals [20]. However, due to the development characteristics of the industry, the macroenvironment, and the international-competition environment, China has not established a comprehensive corporate-social-responsibility evaluation system for forestry companies. Therefore, a reasonable evaluation of corporate social responsibility is not only conducive to the improvement of forestry companies themselves. Competitiveness also positively impacts sustainable development at the social level, such as through environmental protection and low-carbon economy.

## 2. Theory and Method

The definition of corporate social responsibility has always been a fundamental key issue in corporate-social-responsibility research. Therefore, many scholars have conducted indepth research on this issue from different angles, but corporate social responsibility itself is complicated, and they have not come to a unified answer. Currently, some representative definitions are as follows. Clark [21] put forward ideas related to corporate social responsibility as early as 1916. He believed that the core idea of corporate social responsibility is the performance of charitable responsibilities. The definition of the second category of corporate social responsibility is to regard it as society's expectations of corporate behavior. Carroll (1979) [1] showed that corporate social responsibility included the economic, legal, ethical, and discretionary expectations of an economic organization within a certain period of time. This is also known as the "four elements" model. This concept is still widely used. The third definition of corporate social responsibility is to regard it as the compliance of contractual relationships [22]. The labor contracts signed by the company and its employees, the supply and marketing contracts signed with suppliers and customers, etc. are called "contractual relationships" [23]. In order to maintain the fairness of the contract, stakeholders both require the company to aim at maximizing shareholder wealth in production and operation, and to coordinate the interests of all

stakeholders. At present, in the theoretical research of corporate social responsibility, most scholars are currently conducting research on the basis of the "four elements" model of corporate social responsibility by Carroll (1979) [1]. On the basis of existing research, this paper defines corporate social responsibility as the behavior that enterprises should undertake on the basis of profit maximization, targeting shareholders, consumers, the government, the environment, the community, and other stakeholders.

For forestry enterprises, the production and operation process of forestry enterprises involves many factors such as resource management, land ownership, government regulation, and stakeholder management [24]. Because forestry companies rely on their forest resources, their stakeholders and stakeholders' demands for social responsibility for forestry companies also have their own characteristics: consumers expect that the wooden products that they buy are made from sustainable forests, and supplied businesses are more concerned about the price of wood, the protection of primary forests about which government organizations care, and the global carbon cycle, while investors are concerned about the realization of their shareholder value [25]. From the perspective of forestry corporate-social-responsibility behavior, common corporate-social-responsibility behaviors of existing forestry companies include employee, community, ecological, supplier, consumer, and cultural responsibility [26].

For forestry corporate social responsibility, the research focus is mostly on the drivers of corporate social responsibility. Forestry companies' selection of the priority of social-responsibility projects is affected by various factors. Existing analysis of forestry companies' social-responsibility factors is mainly focused on company size and regional country.

(1) Company size. Vidal and Kozak (2008) [27] divided large forestry companies into four categories according to their net sales. Through the study of the relationship between the net sales of large forestry companies and the intensity of social responsibility, they found that the largest forestry companies are engaged in more types of corporate-social-responsibility behavior, medium-sized forestry companies pay more attention to the social and environmental behaviors of corporate social responsibility, and the smallest forestry companies focus on issues such as sustainable forest management. Vidal et al. (2005) [28] found that company size is an important factor that affects whether primary-wood-product manufacturers adopt a regulatory certification system. Large companies were more likely to adopt a regulatory certification system than small companies are, and large companies were more aware of the benefits of adopting a regulatory certification system.

(2) Country area. Political and cultural factors in the location of the company affect corporate-social-responsibility behavior. Panwar et al. (2006) [29] studied the social-responsibility behavior of forestry companies in Europe and the United States, and found that the main driving force for European companies to engage in corporate social responsibility was ethical factors, while the corporate-social-responsibility behavior of American companies was mainly driven by legal factors. Mikkilä (2005) [30] found through research that the social responsibility of forestry companies in Finland, Germany, and Portugal was driven by laws, regulations, and standards, while forestry companies in the Suzhou area of China were voluntary behaviors that engage in social responsibility. In addition, the focus of the social responsibility of forestry enterprises in different regions is also different. Vidal and Kozak (2008) [27] found that there were regional differences in social responsibility. For example, forestry corporate social responsibility in Africa and Latin America focused on social activities. Forestry companies in Asia pay more attention to environmental performance in their business operations; European forestry companies had the most extensive social-responsibility projects, taking into account economic, social, and environmental responsibilities. In addition, the industrial structure, the internationalization level of the industry, the driving of the main stakeholders, the characteristics of CR disclosure, and the attention of the media were all factors that affected the social responsibility of forestry companies [18].

Relative to the number of studies on influencing factors, there are few studies on the evaluation methods of corporate social responsibility. Existing studies mainly use the

analytic hierarchy process, and the fuzzy-comprehensive-evaluation, neural-network-analysis, gray-correlation, and entropy-weight methods for the evaluation of CSR.

(1) The analytic hierarchy process is suitable for situations where it is difficult to directly and accurately measure the results of decision making, but when it is used in CSR evaluation with too many subjective factors, the scaling workload is too large, and it is easy to cause confusion in the judgment of evaluation experts.

(2) The characteristic of the fuzzy-comprehensive-evaluation method is that the evaluation result is not absolutely positive or negative, but expressed by a fuzzy set. Therefore, this method has drawbacks, that is, the evaluator must have a fairly deep understanding of the things being evaluated, especially the knowledge related to CSR.

(3) The neural-network-analysis method finds its rules from complex data through continuous learning, which can better simulate the evaluation process of evaluation experts, but it is difficult to obtain a large number of CSR evaluation training samples.

(4) The gray-comprehensive-evaluation method compares the sequence with reference to the correlation coefficient and correlation degree of the series to determine various influencing factors, and then determines the important factors or the optimal plan, but the comparative series of CSR are difficult to determine due to the characteristics of different industries and time series.

(5) The entropy-weight method is an objective weighting method that uses the amount of information contained in the entropy value of each indicator to determine the weight of the indicator. In recent years, it has been applied to corporate-social-responsibility research. For example, Han and Hanson (2012) [31] established corporate-social-responsibility evaluation indicators for 80 forestry companies on the basis of the entropy method, and found that forestry companies' CSR activities were related to the environment. Related CSR activities were the most, followed by community issues, employee issues, leadership issues, and stakeholder management. Given that the evaluation of CSR in this article is mainly based on the characteristics of objective indicators, using the entropy method to assign weight to indicators can avoid the impact of subjective evaluation, which is more in line with the research purpose and application environment of this article.

However, existing corporate social responsibility has certain defects in the application of entropy law. (1) The quantitative analysis of most numbers is based on the static analysis of corporate-social-responsibility performance without considering time-series factors, the long-term corporate-social-responsibility changes cannot be reflected. This ignores to a certain extent the company's long-term social-responsibility trends. The long-term trend of enterprises in fulfilling social responsibilities can precisely reflect the concepts and attitudes of enterprises to fulfilling social responsibilities. For example, companies that are at a low level of social-responsibility performance in each year do not necessarily lack the concept of undertaking corporate social responsibility, but it may be due to the limitations of the company's scale and profitability. When we take into account the degree of long-term performance of corporate social responsibility in the evaluation elements, similar corporate-social-responsibility trends show an increasing trend. This can separate the company's own management and governance factors, and more clearly reflect its social responsibility concepts and attitudes.

(2) As far as the indicator cross-section is concerned, most evaluation analysis does not subdivide the characteristics of the industry in which the company is located. Taking forestry companies as an example, the focus of social responsibility of forestry companies in different industries is not the same. For example, the social responsibility of the forest-resource service industry focuses on strengthening the protection of forest resources and the construction of forest-park infrastructure. The focus of the social responsibility of afforestation enterprises is to strengthen economic benefits and ecological service functions. Therefore, no matter in which industry category, if companies in different industries are compared in a unified manner, the characteristics of the industry are difficult to uniformly quantify, and obtained results are prone to horizontal deviation. This kind of deviation

may strengthen or weaken some corporate-social-responsibility indicators; thus, quantitative analysis lacks the necessary objectivity and affects the accuracy of the results.

Aiming at the shortcomings of the above-mentioned corporate-social-responsibility evaluation methods, this paper proposes a method to modify the entropy method. This method modifies the entropy method from two levels. On the first level, in order to achieve the comparison of corporate social responsibility in different years, this article adds time variables to the entropy method, that is, to evaluate corporate social responsibility throughout the period. This whole-period evaluation takes into account the changes in corporate social responsibility in each unit period. The second level of correction is the use of relative values. This amendment reflects changes in the performance of the company's social responsibilities in adjacent periods. Therefore, the method in this article can effectively improve the shortcomings of existing corporate-social-responsibility evaluation methods: on the one hand, under this mechanism, companies do not intentionally reduce social responsibility in the early stage in order to achieve a certain year's social-responsibility score. On the other hand, it promotes the fulfillment of corporate social responsibilities and improves social welfare on the basis of corporate social responsibility.

On the basis of the above analysis, this paper selects the data of 17 listed forestry companies for verification in the empirical part. This article selected China's listed forestry companies for three main reasons. (1) In recent years, the continuous attention of all sectors of society to environmental protection has produced a new direction for corporate-social-responsibility research in forestry enterprises. (2) China is a major country in the world's forest-product trade. Forestry is an important part of China's national economy, and plays an important role in economic and social development, and ecological environmental protection. (3) Compared with corporate-social-responsibility research in other industries, scholars have conducted less research on forestry corporate social responsibility, which is in its infancy and has much research space. On this basis, this article selects 17 listed companies in the China Forest Products Processing category in the Shanghai and Shenzhen securities markets as samples. These 17 companies all use forest management and wood as their main raw materials. This paper conducted an empirical evaluation of the performance of social responsibilities of 17 forestry companies from 2011 to 2016. The data were all sourced from the company's annual report. Because different types of enterprises have different foci on fulfilling corporate social responsibilities, the 17 forestry enterprises were divided into 4 types according to the types of forestry business operations: afforestation, wood-processing, furniture-manufacturing, and paper-making enterprises. In addition, the index weights of the corporate social responsibility were calculated in this paper. Corporate-social-responsibility index weights describe the degree of preference of enterprises in the industry to fulfill social responsibility. Different industries have different concerns about the indicators of corporate social responsibility, and index-weight results play a guiding role in the performance of different types of corporate social responsibility.

The main marginal contributions and possible innovations of this article mainly include the three following aspects: First, this article revises the existing corporate-social-responsibility evaluation model and constructs a time-based corporate-social-responsibility evaluation method. This is of great practical significance to help government agencies to better evaluate the performance of corporate social responsibilities. Second, according to the entropy method, this paper carried out a weighted evaluation of the indicators of corporate social responsibility that were studied, and the corporate-social-responsibility ranking could be obtained. Ranking results are conducive to analyzing changes in companies in the process of fulfilling social responsibilities in the long term, guiding companies to better fulfill their social responsibilities, and to the sustainable development of society. Third, this paper used data to conduct social-responsibility evaluation and analysis of listed forestry companies in China. The conclusions of this paper outline the shortcomings of Chinese forestry enterprises in the process of fulfilling their social responsibilities,

and provide theoretical support for Chinese forestry enterprises to fulfill their social responsibilities for the long term.

## 3. Model Building

This article's assessment of CSR is mainly based on data disclosed in the annual reports of each company, that is, objective indicators. Therefore, this article adopted the entropy method to objectively determine the weight of each CSR index. Specifically, by calculating the information entropy of each social-responsibility index of a company, the degree of variation in the corresponding index can be measured; the greater the degree of variation is, the more information is provided by the information entropy, which can be used in the comprehensive evaluation of CSR. The greater the role played, the greater its weight is. By weighting each social-responsibility index, it is possible to accurately examine the degree to which each company fulfills its social responsibility.

On the basis of the entropy method, this article also introduces time-varying factors to measure the degree of change in CSR performance. At present, the entropy method based on the time factor has only been applied to the study of urbanization analysis and land-use efficiency [32], and not to CSR evaluation. This method introduces the time factor into the evaluation of CSR, and fully considers the changes in CSR performance in the past. Therefore, the entropy method based on the time factor does not affect the overall CSR assessment because of the short-term performance of the company. For example, a company ranked first in the industry for CSR in 2016. This indicated that several indicators of CSR were the best in the industry. However, if the long-term performance factors of the company are considered, the relevant indicators of the CSR may show a decreasing trend year by year. Therefore, the 2016 ranking could not fully explain the degree of fulfillment of the CSR. On this basis, we revise the entropy method in the research, which can more comprehensively evaluate corporate social responsibility.

At the specific theoretical level, this article is based on the dynamic model of Xu et al. [33] for application and simulation analysis. Suppose there are $I \in R^+$ companies being evaluated, each company has a one-to-one corresponding quantified social responsibility index $J \in R^+$, and the sample design period is $T \in R^+$ years. Through data disclosed in each annual report of the company, we can obtain quantitative data $x_{ijt}$ on indicator $j \in J$ corresponding to company $i \in I$ in year $t \in T$, and summarize them in original matrix $X_{ijt}$. Through the operation of the original matrix, the weight of each indicator in period T can be determined. The specific steps are as follows:

① Calculate the rate of change.

Because the research on corporate social responsibility in this article includes the changing trends and degrees of each company in the long-term performance of social responsibility, the research method first obtains the change rate of each company's social-responsibility indicators through raw data. Let $r_{ijt}$ be the company $i$'s rate of change for indicator $j$ in year $t$; then, we have

$$r_{ijt} = \frac{x_{ijt} - x_{ij,t-1}}{x_{ij,t-1}}. \tag{1}$$

Summarize the obtained rate of change in matrix $R_{I \times J \times T-1}$.

② Dimensionless processing.

The social-responsibility indicators of each company in the matrix are processed in a dimensionless manner to eliminate the dimensional difference between the indicators. Let $R_{it}^{max}$ and $r_{it}^{min}$ denote the maximal and minimal values, respectively, of the indicators of enterprise $i$ in year $t$; then, we have standardized rate of change

$$s_{ijt} = \begin{cases} \frac{r_{ijt} - r_{it}^{min}}{r_{it}^{max} - r_{it}^{min}}, & if \ r_{ijt} \ is \ positive \ indication \\ \frac{r_{it}^{max} - r_{ijt}}{r_{it}^{max} - r_{it}^{min}}, & if \ r_{ijt} \ is \ negative \ indicator \end{cases}. \tag{2}$$

where positive indicators represent indicators with higher values, such as corporate earnings per share; negative indicators represent indicators with lower values, such as corporate liabilities. Summarize the obtained standardized rate of change in a matrix $S_{I \times J \times T-1}$.

③ Obtain the rate of change throughout the period.

Since the research of this article covers the trend and degree of corporate-social-responsibility performance in the analysis of the whole period, we need to weight the indicators in Formula (2) to obtain the rate of change in the whole period. Here, we chose the weighted-arithmetic-averaging method because it can weaken the influence of the abnormally high or low weight of a company's performance of social responsibility in a certain year. This can help us to investigate as much as possible the overall degree of change in the fulfillment of social responsibilities by each enterprise during the whole period. Therefore, for a company $i$, the rate of change of the company's social-responsibility indicators $j$ over the entire period can be calculated:

$$\bar{s}_{ij} = \frac{\sum_{t=1}^{T-1} s_{ijt}}{T-1} \tag{3}$$

This is summarized in matrix $\overline{S}_{I \times J}$. In this step, we considered the time factor of corporate social responsibility in the data. Therefore, matrix $\overline{S}_{I \times J}$ is already a two-dimensional matrix measuring two dimensions.

④ Normalization processing.

The purpose of normalization is to calculate the weight of each company $i$'s social responsibility on each corporate-social-responsibility indicator $j$. Let the weight of the enterprise on the index be $p_{ij} \in [0,1]$; then, the weight matrix can be expressed as

$$P_{I \times J} = \overline{S}_{I \times J} \otimes \left[ \overline{S}_{I \times J}^{\ T} \times I_{J \times I} \right]^{-1}, \tag{4}$$

where $\overline{S}_{I \times J}^{\ T}$ represents the transposed matrix of matrix $\overline{S}_{I \times J}$, and $I_{J \times I}$ is the identity matrix with dimensions $J \times I$.

⑤ Calculate the entropy and the coefficient of difference.

Summarizing the entropy value of the corporate-social-responsibility index $j$ in the entropy value matrix (column vector) $E_{J \times 1}$, we have

$$E_{J \times 1} = -K \otimes \left[ \left[ P_{I \times J} \otimes ln(P_{I \times J}) \right]^T \times I_{I \times 1} \right], \tag{5}$$

where $K = ln(J)$ is a normal number, and is $I_{I \times 1}$ a column matrix of $I \times 1$. We calculate difference coefficient $d_j$ of corporate-social-responsibility indicators $j$ in the overall sample through the obtained entropy value and summarize them in column matrix $D_{J \times 1}$

$$D_{J \times 1} = I_{J \times 1} - E_{J \times 1} \tag{6}$$

Formula (6) shows that, the smaller the entropy value is, the larger the coefficient of difference, indicating that the social-responsibility index covers more information.

⑥ Calculate the entropy weight.

Calculate the entropy-weight value of index $j$ through difference coefficient $w_j$ obtained in the previous step, and summarize it in entropy weight matrix $\Omega_{J \times 1}$

$$\Omega_{J \times 1} = D_{J \times 1}./\left[ D_{J \times 1}^T \times I_{J \times 1} \right] \tag{7}$$

⑦ Calculate the comprehensive performance of each company's social-responsibility indicators.

Through the weights of each social-responsibility index calculated in Equation (7), we can calculate the weighted value of each enterprise in the performance of social responsibility through the data in matrix $\overline{S}_{I \times J}$, that is, the comprehensive performance quantitative value of each enterprise in fulfilling social responsibility. We use $a_i$ to represent the quantitative value of the comprehensive performance of corporate social responsibility, and summarize it in a matrix $A_{I \times 1}$; then,

$$A_{I\times1} = \overline{S}_{I\times J} \times \Omega_{J\times1} \tag{8}$$

By arranging the various elements in the column matrix, it is possible to examine the overall performance of each company in fulfilling social responsibilities throughout the period.

## 4. Empirical Analysis

### 4.1. Data Description

This paper conducted empirical analysis and evaluation on the CSR of 17 listed Chinese forestry companies in the whole period of 2011–2016. All data come from the annual report released by the companies every year, so the data are objective. According to the management types of forestry companies, the 17 forestry companies were divided into 4 types according to their industries: 2 afforestation companies, 4 wood-processing companies, 2 furniture manufacturing companies, and 9 paper companies. The reason for the classification of forestry enterprises is that different enterprises have different foci on fulfilling social responsibilities. If all forestry enterprises under study were evaluated for social responsibility, the social responsibility weights of different types of enterprises would be confirmed.

On the basis of stakeholder theory proposed by Freeman [34] in 1984, and Yao et al. [35] on the evaluation of forestry corporate social responsibility, all forestry corporate social responsibility was divided into 8 first-level indicators. According to the data availability and quantification of each index, 14 second-level indicators were designed under the first-level indicators (as shown in Table 1). Among them, the secondary index marked (+) represents a positive index, and the secondary index marked (-) represents a negative index.

**Table 1.** Corporate-social-responsibility indicators.

| First-Level Indicators | Second-Level Indicators |
|---|---|
| Responsibility for shareholders | Earnings per share (EPS) (+) |
| | ROE (+) |
| Responsibility for creditors | Current ratio (CR) (+) |
| | Debt asset ratio (DAR) (−) |
| Responsibility for workers | Annual income per employee (AI) (+) |
| | Productivity (+) |
| | Education surcharge (ES) (+) |
| Responsibility for customers | Prime operating revenue (POR) (+) |
| Responsibility for suppliers | Accounts receivable turnover (ART) (+) |
| Responsibility for the government | Tax ratio to main business (TB) (+) |
| | Penalty expense ratio (PER) (−) |
| Responsibility for communities | Donation (+) |
| | Number of paid employees (NE) (+) |
| Responsibility for the environment | Urban maintenance and construction tax (UT) (+) |

Table 2 contains the basic data description of various corporate-social-responsibility indicators in each forestry enterprise category. The table shows that furniture-manufacturing companies had more of the highest average positive index values, while the degree of dispersion of each company between years is higher. Afforestation companies had fewer highest average positive index values, and paper companies had a higher degree of change in social-responsibility performance between enterprises and between years. This shows that, in different forestry industries, the scale of the performance of different social-responsibility indicators by enterprises was not the same. In addition, within the industry,

the extent of the fulfillment of social responsibility by different companies was also significantly different. This shows that, if all forestry enterprises were to conduct a unified corporate-social-responsibility assessment, it would lead to the integration of the rate of change of various indicators and potential information loss. This is also an accuracy-level disadvantage in many current studies that conducted unified social assessments of enterprises in various industries.

**Table 2.** (**a**) Data description of afforestation and wood-processing industries. (**b**) Data description of furniture-making and wood-paper-making industries.

| (a) | | | | | | | | | |
|---|---|---|---|---|---|---|---|---|---|
| | **Firm Type** | **Afforestation** | | | | **Wood Processing** | | | |
| | | **Mean** | **Max** | **Min** | **sd** | **Mean** | **Max** | **Min** | **sd** |
| Responsibility for shareholders | EPS (+) | 0.04 | 0.36 | −0.23 | 0.14 | 0.18 | 1.02 | −0.03 | 0.23 |
| | ROE (+) | 1.51 | 6.88 | −10.96 | 5.26 | 5.37 | 21.22 | −1.94 | 5.33 |
| Responsibility for creditors | CR (+) | 2.45 | 9.79 | 0.77 | 2.82 | 1.13 | 3.67 | 0.37 | 0.83 |
| | DAR (−) | 2.35 | 5.77 | 1.30 | 1.33 | 2.27 | 5.00 | 1.29 | 1.33 |
| Responsibility for workers | AI (+) | 0.58 | 1.97 | 0.11 | 0.53 | 1.07 | 2.69 | 0.22 | 0.66 |
| | Productivity (+) | 52.41 | 73.98 | 21.57 | 16.54 | 65.02 | 138.06 | 15.55 | 34.61 |
| | ES (+) | 228.74 | 920.08 | 49.35 | 250.69 | 687.32 | 2491.83 | 63.02 | 939.37 |
| Responsibility for customers | POR (+) | 7.50 | 15.54 | 3.83 | 3.23 | 28.58 | 84.40 | 3.84 | 29.35 |
| Responsibility for suppliers | ART (+) | 11.78 | 38.09 | 2.82 | 11.40 | 13.57 | 44.96 | 2.83 | 9.60 |
| Responsibility for the government | TB (+) | 0.02 | 0.14 | −0.04 | 0.05 | 0.00 | 0.03 | −0.08 | 0.03 |
| | PER (−) | 0.016% | 0.053% | 0.000% | 0.017% | 0.018% | 0.070% | 0.000% | 0.022% |
| Responsibility for communities | Donation (+) | 10.49 | 31.40 | 0.21 | 9.59 | 52.18 | 450.39 | 0.00 | 117.23 |
| | NE (+) | 1600 | 4133 | 717 | 982 | 4179 | 9085 | 1163 | 2869 |
| Responsibility for the environment | UT (+) | 268.44 | 1083.41 | 72.10 | 290.69 | 975.46 | 3279.63 | 85.95 | 1212.47 |
| (b) | | | | | | | | | |
| | **Firm Type** | **Furniture Making** | | | | **Paper Making** | | | |
| | | **Mean** | **Max** | **Min** | **sd** | **Mean** | **Max** | **Min** | **sd** |
| Responsibility for shareholders | EPS (+) | 0.34 | 0.51 | 0.03 | 0.13 | 0.11 | 0.99 | −0.44 | 0.24 |
| | ROE (+) | 7.95 | 10.58 | 0.83 | 2.63 | 2.24 | 14.12 | −21.26 | 6.01 |
| Responsibility for creditors | CR (+) | 1.77 | 2.69 | 1.13 | 0.44 | 1.07 | 2.20 | 0.29 | 0.45 |
| | DAR (−) | 2.65 | 3.54 | 1.86 | 0.54 | 1.71 | 4.06 | −0.84 | 0.73 |
| Responsibility for workers | AI (+) | 0.81 | 1.67 | 0.22 | 0.41 | 1.18 | 9.51 | 0.06 | 1.60 |
| | Productivity (+) | 55.94 | 77.38 | 42.09 | 13.12 | 107.40 | 221.10 | 38.45 | 48.06 |
| | ES (+) | 1173.87 | 1871.70 | 386.96 | 538.81 | 966.30 | 3551.68 | 120.50 | 920.08 |
| Responsibility for customers | POR (+) | 34.80 | 57.00 | 25.61 | 10.12 | 65.42 | 229.07 | 12.56 | 58.57 |
| Responsibility for suppliers | ART (+) | 5.27 | 7.32 | 3.71 | 0.95 | 8.83 | 19.33 | 3.66 | 3.71 |
| Responsibility for the government | TB (+) | 0.03 | 0.04 | 0.01 | 0.01 | 0.01 | 0.04 | −0.04 | 0.01 |
| | PER (−) | 0.000% | 0.001% | 0.000% | 0.000% | 0.010% | 0.142% | 0.000% | 0.025% |
| Responsibility for communities | Donation (+) | 351.12 | 1041.26 | 33.42 | 308.06 | 26.88 | 280.00 | 0.00 | 43.67 |
| | NE (+) | 6415 | 13,031 | 3928 | 2237 | 5546 | 17,862 | 1561 | 3691 |
| Responsibility for the environment | UT (+) | 1762.60 | 2518.75 | 876.64 | 557.76 | 1342.58 | 5008.47 | 207.63 | 1234.25 |

### 4.2. Results

By using the model introduced in this article to calculate and analyze the quantitative data of forestry corporate social responsibility, it is possible to obtain a ranking of the degree of change in the implementation of social responsibilities of various enterprises during the 2011–2016 period, and the company's performance of various social responsibilities during the whole weight period (i.e., entropy weight). In order to highlight the difference between this method and the conventional entropy method, in this section, we

present the empirical results of the whole year and the empirical results of each year, and explain the meaning and characteristics of its explanation.

### 4.2.1. Corporate-Social-Responsibility Ranking

According to the calculated entropy weight, the social-responsibility indicators of forestry enterprises in the sample were weighted and evaluated, and the ranking of the obtained weighted values corresponds to the social-responsibility ranking of each enterprise. Table 3 shows the corporate-social-responsibility rankings for the whole period based on time factors and the annual corporate-social-responsibility rankings excluding time variables. The former describes the degree of changes in corporate-social-responsibility performance during the whole period of 2011–2016, and the latter describes the degree of corporate-social-responsibility performance in each year. Specifically, the full-period ranking describes the changing trend of corporate social responsibility in different years, while the single-period ranking describes the degree of corporate-social-responsibility fulfillment in a specific year. On the basis of the characteristics of these two rankings, we further analyze the "dynamic" and "static" degrees of companies in different industries in fulfilling their social responsibilities.

**Table 3.** Corporate-social-responsibility ranking.

| Industry | Firm | Rank | Single-Period Rank | | | | | |
|---|---|---|---|---|---|---|---|---|
| | | | 2011 | 2012 | 2013 | 2014 | 2015 | 2016 |
| Afforestation | Pingtan Development | 1 | 1 | 2 | 2 | 2 | 2 | 2 |
| | Yong'an Forestry | 2 | 2 | 1 | 1 | 1 | 1 | 1 |
| Furniture making | Meike Home | 1 | 1 | 2 | 1 | 1 | 1 | 1 |
| | Yihua | 2 | 2 | 1 | 2 | 2 | 2 | 2 |
| Wood processing | Jilin Logging | 1 | 2 | 2 | 2 | 3 | 2 | 4 |
| | Daya Teck | 2 | 4 | 1 | 1 | 1 | 1 | 1 |
| | Shengda Forestry | 3 | 1 | 4 | 4 | 4 | 4 | 3 |
| | Tubaobao | 4 | 3 | 3 | 3 | 2 | 3 | 2 |
| Paper making | Yueyang Paper | 1 | 6 | 2 | 9 | 9 | 3 | 9 |
| | Bohui Paper | 2 | 4 | 9 | 1 | 5 | 8 | 1 |
| | Hengfeng Paper | 3 | 2 | 3 | 2 | 1 | 4 | 2 |
| | SUN Paper | 4 | 3 | 6 | 6 | 2 | 5 | 5 |
| | Chenming Paper | 5 | 8 | 7 | 4 | 3 | 6 | 4 |
| | Qingshan Paper | 6 | 1 | 1 | 7 | 4 | 2 | 7 |
| | Shanying Paper | 7 | 9 | 4 | 5 | 7 | 7 | 8 |
| | Chenxing Paper | 8 | 5 | 8 | 8 | 8 | 9 | 3 |
| | Minfeng Paper | 9 | 7 | 5 | 3 | 6 | 1 | 6 |

Among afforestation enterprises, Pingtan Development's social-responsibility performance in most years was lower than that of Yong'an Forestry. When the time factor was introduced into the inspection indicators, Pingtan Development's weighted value of CSR indicators during the entire period was higher than that of Yong'an Forestry. This shows that, considering the long-term changes in corporate-social-responsibility performance, Pingtan Development had a more obvious increasing trend than that of Yong'an Forestry. Therefore, in Table 3, it appears that companies with a high degree of social-responsibility performance in a single year did not necessarily have a progressive trend of long-term social-responsibility performance. In furniture-manufacturing companies, the social-responsibility rankings of the whole period and the rankings of each year maintained a clear consistency. This shows that, compared with other companies in the industry, Meike Home both had a high degree of social-responsibility fulfillment in each year, and maintained a clear increasing trend in the long term.

The situation of corporate social responsibility in the wood-processing, paper, and paper-product industries was relatively similar. Most of the top-ranked companies in the whole period were in the middle of the single-period social-responsibility performance, while companies that had better corporate-social-responsibility performance in a single period did not show a clear increasing trend. This conclusion reflects that companies with a higher degree of social-responsibility performance in a single year had a more obvious "marginal effect" in the subsequent performance process, that is, although some companies still considered fulfilling social responsibilities, the degree of performance was higher. There was a downward trend before.

### 4.2.2. Weight of Corporate-Social-Responsibility Indicators

Different from corporate-social-responsibility rankings, the weights of corporate-social-responsibility indicators describe the degree of preference for the performance of social responsibilities by enterprise in the industry. The significance of examining the weights of various indicators of corporate social responsibility is to clarify the focus of corporate social responsibility in the long term. This can help us to have a deeper understanding of the corporate-social-responsibility concerns of various industries based on the completion of the corporate-social-responsibility assessment.

As shown in Table 4, different industries have different concerns about corporate-social-responsibility indicators. Specifically, afforestation companies are strongly concerned for shareholder responsibilities in the long term. Among them, the weight of a return on equity and basic earnings per share accounted for 38.81% and 19.94%, respectively, and the total weight was nearly 60%. Regarding afforestation enterprises whose main business is afforestation, due to the protection of forestry resources and strict restrictions on logging that year in China, their main business income has always been at a relatively low level. Table 2 shows that, in the sample industries, afforestation companies had the lowest average value of shareholder responsibility indicators. This has enabled afforestation companies to make adjustments on the basis of their main business to "rely on forest resources, carry out classified management, and expand related extension areas" in order to increase their profits in the operation process. Therefore, the long-term concern of afforestation companies on the level of social responsibility has been implemented in the indicators of shareholder responsibility.

**Table 4.** Social-responsibility index weight.

| First-Level Indicators | Second-Level Indicators | Afforestation | Furniture Making | Wood Processing | Paper Making |
|---|---|---|---|---|---|
| Responsibility for shareholders | Earnings per share (EPS) (+) | 19.94% | 0.58% | 0.65% | 7.04% |
| | Return on equity(ROE) (+) | 38.81% | 0.92% | 1.19% | 7.09% |
| Responsibility for creditors | Current ratio (CR) (+) | 20.91% | 0.98% | 0.63% | 5.44% |
| | Debt asset ratio (DAR) (−) | 2.29% | 0.10% | 1.92% | 9.00% |
| Responsibility for workers | Annual income per employee (AI) (+) | 2.86% | 45.54% | 1.87% | 7.34% |
| | Productivity (+) | 0.05% | 0.03% | 1.37% | 8.07% |
| | Education surcharge (ES) (+) | 0.14% | 8.01% | 2.16% | 6.23% |
| Responsibility for customers | Prime operating revenue (POR) (+) | 1.12% | 0.36% | 1.83% | 7.18% |
| Responsibility for suppliers | Accounts receivable turnover (ART) (+) | 3.03% | 4.75% | 1.91% | 5.62% |
| Responsibility for the government | Tax ratio to main business (TB) (+) | 1.85% | 16.10% | 1.79% | 7.80% |
| | Penalty expense ratio (PER) (−) | 1.43% | 0.05% | 79.69% | 9.01% |
| Responsibility for communities | Donation (+) | 2.44% | 13.82% | 0.11% | 8.31% |
| | Number of paid employees (NE) (+) | 5.06% | 1.20% | 2.78% | 5.74% |
| Responsibility for the environment | Urban maintenance and construction tax (UT) (+) | 0.06% | 7.55% | 2.10% | 6.14% |
| | Mean | 6.16% | 7.65% | 7.64% | 7.15% |
| | *p* value | 0.0678 | 0.0763 | 0.1309 | 0.0075 |
| | sd | 10.98% | 11.82% | 20.13% | 1.15% |

In the furniture-manufacturing industry, the focus of corporate social responsibility lies in employee responsibility. The data in Table 2 show that the furniture-manufacturing industry has the most employees. Therefore, the degree of concern for employee welfare is an important practice for most furniture-manufacturing enterprises' social responsibility. Since the evaluation method used in this article covers long-term changes to various indicators, furniture-manufacturing companies attach a very high degree of importance to employee salary increments.

Table 4 shows that companies in the wood-processing, paper, and paper-product industries have paid more attention to their responsibilities to the government in the long term, especially for a reduction in fines. Reducing fines can be seen as a way to regulate operations and reduce violations by fulfilling social responsibilities. Among them, the wood-processing industry uses wood as raw material to produce wood products through various chemical-liquid treatments or mechanical-processing methods, such as wood-based panel manufacturing, wood-based panel decoration treatment, and wood chemical treatment; for the paper-making and -product industries, most processes, such as pulping, alkali recovery, and bleaching produce waste water, waste gas, waste residue, and toxic substances, causing serious pollution to the environment. Therefore, the government's rules for these two types of enterprises are particularly strict, which makes the enterprises attach great importance to standardized operations in the long run. In addition, Table 4 reflects that paper and paper-product companies do not deviate much from various social-responsibility indicators, that is, the standard deviation of social-responsibility indicators in the industry is only 1.15%. This also shows that paper and paper-product enterprises have paid obvious attention to various indicators in the process of the long-term fulfillment of social responsibilities, and did not focus their attention on certain indicators.

## 5. Conclusions and Policy Implications

This paper examined the degree of long-term performance of corporate social responsibility by enterprises, incorporated the degree of change in performing social responsibility into the category of social-responsibility evaluation, and constructed a time-based corporate-social-responsibility evaluation method. The assessment of the social responsibility of 17 listed forestry companies in China showed that the time factor significantly impacts the ranking of corporate social responsibility. Specifically, companies with a high degree of the fulfillment of social responsibilities in a single period do not necessarily have an increasing trend in the completion of social responsibilities in the long term. This shows that the evaluation of corporate social responsibility should not only consider the company's fulfillment of social responsibilities in a single period. The degree of change in the company's long-term social responsibility should also be considered. In addition, research in this article provides the long-term focus of companies in different industries in fulfilling their social responsibilities. Different types of enterprises place different emphasis on social-responsibility indicators on the basis of their own characteristics. This provides a clearer information path for the identification of corporate-social-responsibility standards, norms, and constraints in different industries, and the development focus of enterprises in the industry. The research in this paper guides enterprises to better fulfill their social responsibilities, is conducive to the sustainable development of society, presents the shortcomings of Chinese forestry enterprises in fulfilling their social responsibilities, and provides theoretical support for Chinese forestry enterprises to fulfill their social responsibilities for the long term.

As an empirical study based on the revision of corporate-social-responsibility evaluation theory, this article still has the following shortcomings. First, the data used in this article only come from listed forestry companies in China and do not examine companies in other industries. This also puts forward new requirements for subsequent research. Second, research in this article does not cover the unique events of each company during the inspection period, such as corporate restructuring and major projects launching. This is mainly because such indicators are not easy to quantify, but also leads to the lack of information. Therefore, in subsequent research, we aim to focus on exploring a set of scientific quantitative methods in order to minimize the lack of CSR assessment information.

**Author Contributions:** X.L. and H.X., calculation, data analysis, and paper writing; X.L. and B.C., argument verification and solidification, paper editing, and conclusion drafting; X.L., manuscript revision during the whole writing process. All authors have read and agreed to the published version of the manuscript.

**Funding:** National Natural Science Foundation of China (72073012; 71873016).

**Institutional Review Board Statement:** Not applicable.

**Informed Consent Statement:** Not applicable.

**Data Availability Statement:** Data were selected from the database of Tai'an at https://www.gtarsc.com/.

**Conflicts of Interest:** The authors declare no conflict of interest.

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
