# Peer review of "Time-Based Corporate-Social-Responsibility Evaluation Model Taking Chinese Listed Forestry Companies as an Example"

_sustainability, doi:10.3390/su13147971_

Round 1
Reviewer 1 Report
This study examines a time-based social responsibility evaluation model using Chinese listed forestry companies. The results of the study suggested that the long-term trend of CSR may not fully reflect the static evaluation (fixed evaluation) of CSR.
It is difficult to generalize the research results because the research and analysis targets only forestry companies, not the overall companies. It is also thought that the research results are less applicable because it is targeted at only a small number of forestry enterprises. Therefore, it is judged that the difference is not significant compared to the previous study.
Please fully check the suggestions below and reflect them in your paper.
- Differentiation and Contribution Points from Prior Research
In this study, CSR is argued as one of the important factors for companies to strengthen their market competitiveness in a competitive environment. As CSR has become a clear legal obligation in China since 2006, this study is considered an important research topic under the socialist system. However, this paper did not look at all listed companies in China, but only 17 forest products processing companies in the Shanghai and Shenzhen stock markets were studied. It is thought that the analysis results will depend greatly on the characteristics of several companies. Therefore, it is difficult to generalize the research results. It also describes very simple reasons for choosing forestry enterprise data in China. Because CSR is important to other companies as well, there is a need to study the industry as a whole, not just forestry. I think this study is only part of the industry-specific additional analysis because it is not based on the entire company. Therefore, rather than limiting the research target company to forestry, it is thought that the contribution of this study will be significant by analyzing the entire company and then further examining the effects of each industry. At this point, only 17 forestry processing companies are analyzed as research subjects, so this study is judged to have no contribution and difference compared to prior research. After examining the overall Chinese companies, it is necessary to further compare the types of industries to clearly understand the differences.
- There is a need to clarify the part related to hypothesis setting
Only when the parts related to the hypothesis setting that this study wants to claim are separately organized and sufficiently described, can the contents of this study be clearly explained. At this point, this study is conducting an analysis after describing a prior study related to CSR in the introduction section. Please specify the necessity of this study and the specific reason for conducting this study.
Reviewer 2 Report
The abstract well summarizes the content and the aim of the paper.
The theoretical background and the introduction section should be divided into two paragraphs.
The authors could clarify the research gaps that emerge and how they want to fill it.
Due to the relevance of CSR topic (not only its evaluation), the authors could frame the theme under the perspective of stakeholder theory (Freeman, 1984). In particular, it could be investigated how forestry firms implement CSR activities. Furthermore, in the context analysed, the sustainability theme is of greater relevance too.
Results are explained in a clearly way.
The study is lacking of practical implications.
Reviewer 3 Report
The quantitative evalutation of Corporate Social Responsibility policies is an interesting topic, however I believe that the manuscript has several weaknesses that should be addressed.
1) The literature is quite limited, both - and above all - in reference to the concept of CSR, and to the more specific literature on evaluation methods. The only reported definition is that of Carroll, of 1979. McWilliams and Siegel are also cited, but the conceptualization and contestualization are really poor: the background should better frame, albeit in summary, 50 years of CSR literature or the authors should clearly define which strand of literature they refer to and therefore which CSR perspective they want to adopt. For example, it seems that they refer exclusively to a strategic and instrumental perspective of CSR, not ethics. The literature on the CSR evaluation methods could also be improved and expanded. The paragraph would benefit from a different organization (e.g. into two paragraphs), in the actual version it seems to have little internal coherence.
2) Does the lines 81-92 express a personal opinion of the authors? The statements, even important ones, inserted do not contain references to the literature, nor do the authors explain how they reached these conclusions. Moreover, the phrase "the literatures mainly include qualitative analysis methods and quantitative analysis method" is really difficult to understand: are there other types of methods? Maybe, it is meant that mixed-approach studies are lacking? Please, clarify.
3) Authors seem to assert that quantitative methods are always better than qualitative methods for evaluation.
4) Lines 177-197 sound more as a conclusion than an introduction. In any case, the originality could be better - but more succinctly - highlighted.
5) The overall investigated sector and the sub-group 1 have the same denomination. Please, recheck.
6) Table 1: These indicators seems more economic and less social/environmental. Please, clarify the reasons for this choice (the clarification of CSR concept could help).
7) Conclusion/discussion are not linked with previous literature. In addition, implications and contribution should be improved.
8) At line 442, authors wrote "in my country". Please, recheck.
Round 2
Reviewer 1 Report
Congratulations on this revision. Nice work.